# Dietary Preference of Newly Weaned Pigs and Nutrient Interactions According to Copper Levels and Sources with Different Solubility Characteristics

**DOI:** 10.3390/ani10071133

**Published:** 2020-07-03

**Authors:** Sandra Villagómez-Estrada, José Francisco Pérez, Sandra van Kuijk, Diego Melo-Durán, Razzagh Karimirad, David Solà-Oriol

**Affiliations:** 1Animal Nutrition and Welfare Service (SNIBA), Department of Animal and Food Science, Universitat Autonòma de Barcelona, 08193 Bellaterra, Spain; sandra.villagomez@outlook.es (S.V.-E.); josefrancisco.perez@uab.cat (J.F.P.); Diego.Melo@uab.cat (D.M.-D.); karimiyashar6@gmail.com (R.K.); 2Trouw Nutrition, Research and Development Department, 3800 Amersfoort, The Netherlands; sandra.van.kuijk@trouwnutrition.com; 3Department of Animal Science, Lorestan University, 68137-17133 Khorramabad, Iran

**Keywords:** copper, feed preference, levels, post-ingestive effect, sources, solubility

## Abstract

**Simple Summary:**

Strategies for promoting early feed acceptance and avoiding pig rejection to new feed are a priority for maximizing their feed intake. Animal preference or aversion for a particular feed or nutrient is a behavioral expression coordinated by a complex biological system. After weaning, Cu blood level decrease, which is probably intensified by a low feed intake. This can lead to suboptimal Cu level for the normal functioning of the body. In the present study two experiments were performed to assess the pig Cu preference. In Exp.1 (dose preference) pigs were given a choice between diets supplemented with Cu at 15 mg/kg or 150 mg/kg. In Exp.2 (source preference) diets supplemented with Cu at 150 mg/kg were offered with either sulfate or hydroxychloride source. An in vitro assay was performed to determine the Cu solubility of each source in similar conditions to those found in the oral cavity and digesta. Our results show that pigs chose diets with higher Cu levels, probably to re-establish homeostasis after weaning. Pigs preferred diets with Cu hydroxychloride compared to Cu sulfate, probably due to their solubility differences. A better understanding of pig feed preferences after weaning and their feeding behavior would improve early feed acceptance.

**Abstract:**

Two feeding preference experiments and an in vitro assay were performed to assess the weaned pig preference for Cu doses and sources based on their sensorial perception and on the likely post-ingestive effects of Cu. At day 7 post-weaning, a total of 828 pigs were distributed into two different experiments. In Exp.1 (dose preference) a diet with a nutritional Cu level (15 mg/kg) of Cu sulfate (SF) was pair offered with higher Cu levels (150 mg/kg) of either SF or hydroxychloride (HCl). In Exp.2 (source preference), a diet supplemented with Cu-SF at 150 mg/kg was compared to a Cu-HCl (150 mg/kg) diet. At the short-term (day 7–9) and for the entire experimental week (day 7–14), pigs preferred diets with a high Cu level than with Cu at a nutritional dose (*p* < 0.05). Likewise, pigs preferred diets supplemented with a Cu-HCl source compared to diets with Cu-SF (*p* < 0.05). In vitro assay results showed a greater solubility and interaction of Cu-SF with phytic acid compared to Cu-HCl. In conclusion, pigs chose diets with higher levels of Cu probably to re-establish homeostasis after weaning. Pigs preferred diets with Cu-HCl compared to Cu-SF probably due to their solubilities and chemical differences.

## 1. Introduction

Weaning is considered the most critical period in swine production due to low feed intake, gastrointestinal disturbances, impaired gut integrity and consequently growth. Strategies for promoting early feed acceptance and avoiding feed neophobia and rejection after weaning are a priority for maximizing feed intake.

Animal preference or aversion for a particular feed or nutrient is a behavioral expression orchestrated by a complex biological system that involves different organs and sensory, metabolic and physiological feedbacks [1,2]. Several studies have confirmed that pigs have the ability to select among dietary ingredients and/or nutrients, for example, types of cereals [3] and amino acids [4]. Pigs have also been found to modulate their feed preference under nutrient deficit scenarios in order to re-establish homeostasis [5,6]. In terms of mineral nutrition, the swine industry usually includes high quantities of trace minerals such as Cu or Zn in pig diets due to their beneficial effects for controlling gastrointestinal dysbiosis and increasing feed intake [7]. To our knowledge, only one previous study has assessed the preference of finishing pigs (body weight; BW: 86 kg, approximately) for diets without Cu supplementation or diets with Cu at 150 mg/kg [8]. After weaning, blood levels of some trace minerals, such as Cu (1.8 mg/L vs. 1.4 mg/L) and Zn (1.10 mg/L vs. 0.76 mg/L), may decrease [9,10]. This, together with the low feed intake after weaning, may lead to suboptimal Cu and Zn levels for a proper metabolic and immune response. Since the preference for a diet may also be influenced by the animals’ nutritional requirements [6,11], it is relevant to gain a better understanding of weaned pigs’ preferences and acceptance for trace minerals within the range of nutritional and high inclusion levels, as well as different commercial sources.

Studies with humans have concluded that the taste of Cu in water depends on its chemical structures and its solubility [12,13]. Since pigs have approximately three times more taste buds than humans [14], it is likely that pigs have superior taste perception. Cu sulfate pentahydrate is a widely used source in pig diets. It is characterized by a high solubility in water and acid solutions and hence higher interactions with other components of the diet, including phytic acid and other minerals [15,16,17]. Phytic acid derived from plant ingredients can easily bind divalent metal ions such as Cu, resulting in mineral–phytin complexes and reducing the solubility of Cu and P [16]. Another common Cu source is Cu hydroxychloride. This has a crystalline structure formed by covalent bonds that gives it low solubility above pH 4 and insolubility in water [16].

The present study was carried out to test the hypothesis that weaned pigs will be able to select feed based on the sensorial perception of Cu sources due to differences in their solubility, and that pigs will choose feed based on the likely post-ingestive effects of Cu, resulting in a preference or aversion for feed. We also hypothesized that the effective solubility parameters will also depend on the interaction between Cu and phytic acid, as these are different among Cu sources. Thus, the aim of the study was to evaluate the preference of weaned pigs for diets that contain added Cu at 15 mg/kg (nutritional) and 150 mg/kg (high) as inclusion levels (dose preference experiment, Experiment (1) from the source of either Cu sulfate or Cu hydroxychloride (source preference experiment, Experiment (2). In addition, an in vitro assay was performed to determine trace mineral properties in similar conditions to those found in the oral cavity and digestive tract by testing in vitro the Cu solubility of each source (at 50, 100, 200 and 300 mg Cu/L) and its interaction with phytic acid, in buffer solutions at pH 2.5, 4.5 and 6.5. Together, the animal feed experiments, and the in vitro assay are intended to describe some clues of pig preference for different Cu levels (15 vs. 150 mg/kg) and sources (sulfate vs. hydroxychloride). In the establishment of the methodology of the present study some points were considered, such as a suitably methodology for evaluate animal preference, measurement points, length of phases and weaning age of animals under commercial conditions, as well as the simultaneous comparison between nutritional and high Cu doses in newly weaned pigs. All these considerations make the present study different from previous related works on mineral nutrition.

## 2. Materials and Methods

### 2.1. Ethics Statement

All animal experimentation procedures were approved by the Ethics Committee of the Universitat Autònoma de Barcelona in compliance with the European Union guidelines for the care and use of animals in research (approval code CEEAH2788M2) [18].

### 2.2. Feed Preference Experiment (Experiment 1 and 2)

#### 2.2.1. Animals and Housing

The present preference experiments were performed under Spanish commercial conditions. A total of 828 entire male and female pigs ((Large White × Landrace) × Pietrain) were selected to be used in two different preference experiments. Pigs were weaned at day 21 of age and housed in a weanling room belonging to the same commercial farm. At day 7 after weaning, pigs were blocked by sex and distributed, according to a homogenous BW, into different experimental treatments. In the first double choice test (dose preference), 552 pigs (initial BW: 6.6 ± 1.10 kg) were allocated to 24 pens (23 pigs/pen) randomly assigned to two different treatments (12 replicate pens/treatment). In the second double choice test (source preference), 276 pigs (initial BW: 7.1 ± 1.08 kg) were allocated to 12 pens (23 pigs/pen/treatment). Pens of males and females were assigned equally to dietary treatments. Each pen (4.04 m^2^) was equipped with two commercial pan feeders (Maxi hopper, Rotecna, Spain) and a nipple drinker to provide ad libitum access to feed and water. The facility was environmentally controlled (temperature and ventilation rate) using thermostatically controlled heaters and exhaust fans.

#### 2.2.2. Experimental Design and Dietary Treatments

In order to avoid biases due to the low feed consumption of newly weaned pigs during the first week after weaning (day 1–7 post-weaning), pigs were adapted to the new environmental commercial conditions by offering ad libitum a commercial pre-starter diet including 9 mg Cu/kg. The preference test was performed during the second week (day 7–14) after weaning. At day 7 post-weaning, pan hopper feeders used during the adaptation phase were completely emptied and replaced by two new pan hopper feeders to provide the reference diet and the assigned experimental diet in each pen. In order to avoid biases, both pan hopper feeders were hand filled to ensure completely free access to the two diets. The position of the feeders (right or left) inside the pen was switched once (day 9) to prevent location bias in feeding behavior.

In the dose preference test, a diet with a nutritional Cu level (15 mg Cu/kg) of Cu sulfate pentahydrate (Pintaluba, Reus, Spain) was used as a reference diet and was pair offered with higher Cu levels (150 mg/kg) of either Cu sulfate or Cu hydroxychloride (IntelliBond C, Trouw Nutrition, the Netherlands). In contrast, in the source preference test, the reference diet was supplemented with Cu (150 mg/kg) as sulfate pentahydrate and compared to a Cu hydroxychloride (150 mg/kg) diet. Supplementation of Zn (Sulfate monohydrate; Pintaluba, Reus, Spain) was fixed for all diets at 100 mg/kg, according to National Research Council (NRC) [19] requirements. Feed was offered in pellet form and both feed and water were provided ad libitum. A vitamin–mineral premix without Cu was prepared. For each dietary treatment, Cu products were pre-mixed with 25 kg of basal diet before being put directly in the mixer during the feed preparation process. The composition of the basal diet is shown in Table 1. Diets contained 500 units of phytase (Axtra PHY TPT, Danisco, Marlborough, UK) per kg of complete feed.

Composite samples (1 kg) were collected during the bagging process as representations of each experimental treatment. Each sample was therefore split proportionally into four 250 g samples that were stored for further analysis. Zinc oxide was not added at pharmacological levels to the diets and no antibiotics or feed additives with flavoring or antimicrobial properties were used.

#### 2.2.3. Experimental Procedures and Measurements

Feed intake for the reference and the experimental diets was measured from day 7–9 (short-term preference), and from day 9–14 (long-term preference). The methodology and reference periods were defined from the works of Forbes [6], Torrallardona and Solà-Oriol [20] and Roura et al. [11]. Thus, the short-term preference (2 d in the present study) is considered as the immediate response of animals to the different sensory characteristics of the feed, whereas the long-term preference (5 day in the present study) might be defined as more than a few days determined by learned preferences and aversions according the metabolic response of animals to the feed. The feed intake values of each diet were expressed as described in [21] but with a modification. Briefly, feed intake per replicate pen was standardized by dividing the diet feed intake by the average pig body weight and by the number of pigs per replicate pen. Then, the preference of the test diet relative to the reference diet was calculated as the percentage contribution of the test diet to the total feed intake as described in [3]. Preference values can range between 100 and 0%. A value of 50% would indicate indifference with respect to the reference diet, whereas values significantly higher or lower than 50% would indicate a significant preference or aversion, respectively.

### 2.3. In Vitro Assay

The solubility of Cu from a sulfate or hydroxychloride source and the concentration of soluble phytic phosphorus (PP) were measured in duplicate. Cu was added at concentrations of 50 mg/L, 100 mg/L, 200 mg/L and 300 mg/L in 200 mM glycine buffer (pH 2.5) (Merck, Germany) and 200 mM sodium acetate buffer (pH 4.5 and 6.5) (Merck, Germany). Each Cu source was mixed with 20 mL of buffer with and without 2.9 mM phytic acid (Merck, Germany), incubated at 41 °C in a shaking water bath for 1 h and filtered through 42 μm Whatman filter paper. Soluble Cu and PP analysis was carried out with inductively coupled plasma-optical emission spectroscopy (ICP-OES, model Optima 4300 DV, PerkinElmer Inc.; Waltham, MA, US). The solubility of PP was expressed as mg/L, whereas the solubility of Cu was calculated using the following equation (1):Mineral Solubility (%) = (Soluble mineral)/(Total mineral) × 100(1)

### 2.4. Statistical Analysis

Standardized preference percentage values were analyzed as a randomized complete block design using the MIXED procedure of SAS (version 9.4, SAS Institute; Cary, US). The model included the fixed effects of treatment and the random effects of sex. The pen was considered the experimental unit. The normality and homogeneity of the data were examined with the Shapiro–Wilk statistical test of SAS^®^ before statistical analysis. In addition, the preference percentage values of each experimental diet were compared to the neutral value of 50% using a Student’s T-test procedure. Significantly different means were separated using the Tukey adjust. Significance was defined at a probability *p* ≤ 0.05 and tendencies were considered when *p*-values were between >0.05 and <0.10.

## 3. Results

The analyzed mineral concentrations in feed are shown in Table 1. The expected difference between nutritional and high levels of Cu in the diets was achieved with both sulfate and hydroxychloride Cu sources.

The growth performance of pigs for preference experiments are shown in Table 2. Since the two experimental diets were provided at the same time in each pen, differences in growth performance cannot be attributed to either diet.

### 3.1. Dose Preference Test (Exp. 1)

The pig preference response for the experimental diets is shown in Table 3. In the short-term, when pigs were given a choice between diets supplemented with nutritional or high Cu level, they preferred high Cu diets, regardless of the source (*p* < 0.05). In the long-term period, the feed preference was not different between experimental diets (*p* < 0.10). Considering the entire experimental period, pigs preferred diets supplemented with high Cu level with a hydroxychloride source than those supplemented with a nutritional Cu level (58.9% vs. 41.1%; *p* = 0.035).

### 3.2. Source Preference Test (Exp. 2)

Pig preference for Cu sources is shown in Table 3. In the short-term, pigs showed a greater preference for diets supplemented with the Cu hydroxychloride source than those with the Cu sulfate (62.2% vs. 37.8%; *p* = 0.004). In the long-term period, no differences were observed (*p* = 0.141). Considering the entire experimental period, pigs preferred diets supplemented with Cu hydroxychloride compared to those supplemented with Cu sulfate (57.3% vs. 42.7%; *p* = 0.039).

### 3.3. In Vitro Assay

The results of Cu solubility and interaction with phytic acid of the two Cu sources are shown in Figure 1, Figure 2 and Figure 3. As shown in Figure 1, the mean Cu solubility of Cu sulfate, as an average of the four concentrations, was 98% in the whole range of pH levels (2.5–6.5), whereas the hydroxychloride Cu solubility, as an average of the four concentrations, decreased drastically from 100% at pH 2.5 and 4.5 to 9% at pH 6.5.

Overall, the presence of phytic acid decreased the Cu solubility as pH increased (from 99% at pH 2.5 to 21% at pH 6.5; Figure 2 and Figure 3), regardless of the source. At pH 2.5, the solubility of Cu was similar for both sources and it was not modified by the presence of phytic acid in samples (Figure 2).

At pH 4.5 the addition of phytic acid decreased the mean Cu solubility of Cu sulfate from 100% (without phytic acid) to 88% (Figure 3a). Likewise, the mean Cu solubility of Cu hydroxychloride decreased from 100% to 56% when phytic acid was added (Figure 3a). At intestinal pH (6.5), the presence of phytic acid did not affect Cu solubility of Cu hydroxychloride but sharply reduced Cu solubility of Cu sulfate (Figure 3b).

In general, soluble PP decreased as the pH and the Cu dose increased (Figure 2 and Figure 3). At pH 2.5 and 4.5 no substantial differences between the two Cu sources were observed in soluble PP content (Figure 2 and Figure 3a), whereas at pH 6.5, increasing levels of Cu with a sulfate source, but not with a hydroxychloride source, notably reduced the content of soluble PP (Figure 3b).

## 4. Discussion

The average daily feed intake (ADFI) of pigs during the experimental week was within the commercial standard values, and were 201 g and 204 g for the dose and source preference studies, respectively. One of the proposed hypotheses of the present study was that weaned pigs might be able to select feed based on the likely post-ingestive effects of Cu on pigs. Our results showed that when pigs are given a choice, they preferred diets supplemented with higher Cu levels over those with nutritional levels.

In pig life, weaning is considered one of the most difficult phases. Two phases can be distinguished after weaning based on the changes in feed intake and the subsequent impacts on the physiology of the gastrointestinal tract: an acute phase (within the first 5–7 days after weaning) and an adaptive phase (day 7 after weaning and onwards) [22]. The temporary low feed intake (below 100 g/day) during the acute phase may create a deficiency in macronutrients, micronutrients and a negative energy balance, which can impair health, development and recovery during the adaptive phase [23,24]. For instance, Carlson et al. [7] reported that Cu plasma status decreased during the first two weeks after weaning, from 1.8 mg/L to 1.4 mg/L. The NRC [19] established a minimum requirement of Cu (6 mg) per kg of diet or as a daily Cu requirement (1.6–2.8 mg) according to the BW (5–11 kg) of pigs. During the lactation period, it is assumed that the consumption of milk meets the suckling pigs’ daily Cu physiological requirement. However, to meet this Cu requirement after weaning, pigs must consume 267 g feed/day if feed contains 6 mg/kg as recommended by the NRC. As mentioned before, after pigs are weaned, they have an extremely low feed intake, which may promote this post-weaning Cu deficiency. In the present study during the first week after weaning, pigs were offered a commercial diet with Cu supplementation (9 mg/kg) above NRC recommendations [19], but it was probably insufficient during the critical first period after weaning. Although the serum Cu status of pigs after weaning was not measured, our results suggest that pigs’ preference for diets supplemented with high levels of Cu is probably a systemic response to restore Cu levels after weaning. In this sense, Bikker et al. [7] reported that after 56 days of supplementation, pigs fed low Cu diets (15 mg/kg) had higher mRNA levels of Cu transporters in their upper small intestine than those fed high Cu levels (160 mg/kg), probably as a compensatory mechanism to maintain Cu homeostasis. Roura et al. [11,25] pointed out that the preference mechanisms of an animal for a feed are mainly determined by chemosensory signals from the upper gastrointestinal tract (mouth and stomach; short term) and by the integration of post-gastric signals (from small or large intestines; long-term) to the brain. Thus, the nutritional value of the food can be distinguished and translated into a physiological stimulus that ends up triggering a greater or lesser consumption of food [11,25]. Previous studies have shown that animals are able to adjust their feed intake based on nutrient requirements, the nutritional value of diets, and health status [6,11]. Concerning minerals, animals deficient in certain minerals such as Na, Ca or P have been found to be able to select a food supplemented with the required nutrient [26,27]. In our laboratory, the same feeding behavior pattern was observed with other nutrients, such as protein. Guzmán-Pino et al. [5] reported that weaned pigs were able to choose feed to correct underfeeding or a protein deficient status and thus re-establish homeostasis. In contrast, Coble et al. [8] reported that finishing pigs had a higher preference for diets without supplementary Cu than those supplemented with 150 mg/Cu kg. It must be noted that unlike weaned pigs, finishing pigs have already overcome the temporary physiological Cu deficit and gastrointestinal stress produced by weaning. Therefore, the preference of an animal for a certain feed may be more than just a matter of flavor perception [28], and probably reflects an interrelationship between the perception of the flavor of the feed and its post-ingestive effects [11]. Among all the positive effects attributed to Cu supplementation, Bikker et al. [7] concluded that approximately 75% of the growth promotor effect of Cu in weaned pigs is explained by the increase in feed intake. The authors also showed that the linear supplementation of Cu (15, 80, 120 and 160 mg/kg) in weaned pig diets resulted in a linear improvement in growth performance. Since in the present study pigs were offered nutritional and high doses of Cu at the same time, no distinctions in growth performance could be established. However, in a previous study performed in our laboratory, the supplementation of weanling pig diets with 160 mg Cu/kg increased ADFI (360 to 379 g), gain to feed ratio and BW(16.6 to 17.7 kg) compared to diets with 15 mg/Cu kg after 42 days of administration (*p* < 0.05) [29].

When Cu sources were contrasted at 150 mg Cu/kg, a large preference was observed for Cu hydroxychloride compared to Cu sulfate. Based on human studies, it is known that the taste of Cu in water depends on its chemical structures and solubility, and the taste has been described as metallic, bitter and bloody [12,13]. Since, a pig’s tongue has approximately three times more taste buds involved in the complex process of diet selection compared to humans [14], it is likely that taste perception in pigs is superior to that of humans. Cuppett et al. [13] showed that free Cu ions and soluble Cu complexes in a pH range of from 6.5 to 7.4 can be readily tasted in water. The results of the present in vitro assay showed that Cu hydroxychloride solubility decreased from 100% at pH 2.5 to 9% at pH 6.5 (approximate oral cavity and intestinal pH), unlike Cu sulfate (99–98%, respectively). In agreement with this, Pang and Applegate [30] reported that the order of the solubility of three Cu sources at 250 mg/kg was Cu sulfate > Cu lysine > Cu hydroxychloride for each pH measured (pH 2.5, 5.5, and 6.5). Therefore, it could be speculated that the taste perception of pigs is more intense when Cu sulfate is added compared to Cu hydroxychloride, mainly due to its solubility characteristics. Early studies with finishing pigs reported a higher preference for diets supplemented with Cu at 150 mg/kg as hydroxychloride (65% vs. 35%) than diets with Cu sulfate over 15 days [8]. Likewise, when weaned pigs were offered diets supplemented with Cu at 160 mg/kg as hydroxychloride or sulfate, a strong preference was observed for hydroxychloride compared to sulfate during the first two weeks post-weaning (76% vs. 24% and 81% vs. 19%, respectively) [31].

In the present in vitro assay, Cu solubility was affected by the addition of phytic acid. The largest inhibition of Cu solubility was observed at pH 6.5, that is, Cu sulfate inhibited more intensely than Cu hydroxychloride. Likewise, soluble PP concentrations were affected by the increase in Cu doses and were more critical at pH 6.5 and with the Cu sulfate source. High levels of Cu as sulfate have been described to interfere with phytate, especially at intestinal pH (6.5), resulting in chelate complexes that tend to be resistant to the hydrolytic activity of phytases [13,14]. In 2006, Pang and Applegate [16] in a comparative study reported that adding higher Cu concentrations inhibited PP hydrolysis by phytase at pH 5.5 and 6.5, while Cu as hydroxychloride and lysinate inhibited PP hydrolysis much less than Cu as sulfate pentahydrate, chloride or citrate. Once more, these differences may be the result of the different solubilities among Cu sources.

## 5. Conclusions

In conclusion, pigs’ preference for an essential nutrient like Cu responds to a more complex physiological mechanism than just a flavor perception. Weaned pigs were able to choose diets supplemented with higher levels of Cu, probably due to its systemic post-ingestive effect in order to re-establish Cu homeostasis after weaning. Pigs’ Cu taste perception between sources is also attributed to the solubilities and chemical differences of the sources. Pigs showed a greater preference for Cu hydroxychloride diets than those containing the more soluble Cu sulfate. Complementary studies using new approaches, such as genomic tools, should be conducted to explore the biological mechanism behind Cu preference at this stage.

## Figures and Tables

**Figure 1 animals-10-01133-f001:**
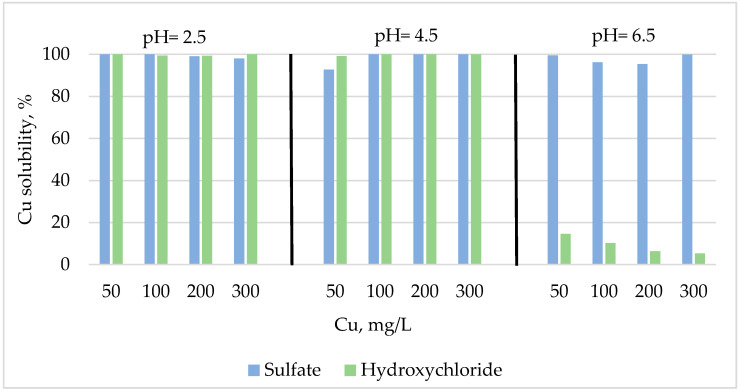
Effects of Cu source and level on Cu solubility at pH 2.5, 4.5, and 6.5 in the absence of phytic acid.

**Figure 2 animals-10-01133-f002:**
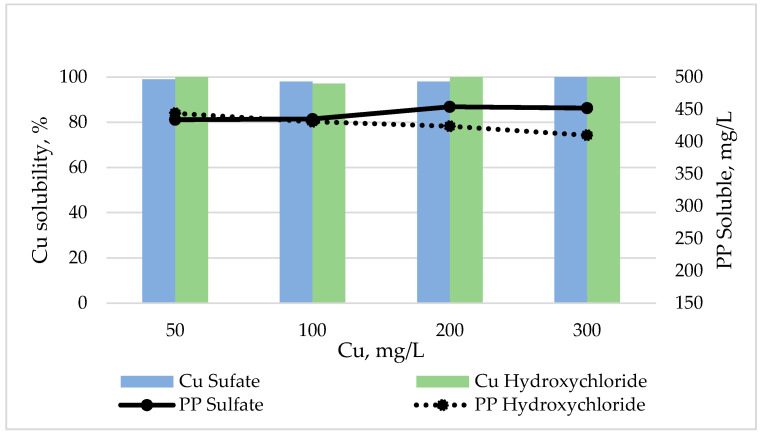
Effects of Cu source and level on Cu solubility and soluble phytic phosphorus (PP) content at pH 2.5 in the presence of phytic acid.

**Figure 3 animals-10-01133-f003:**
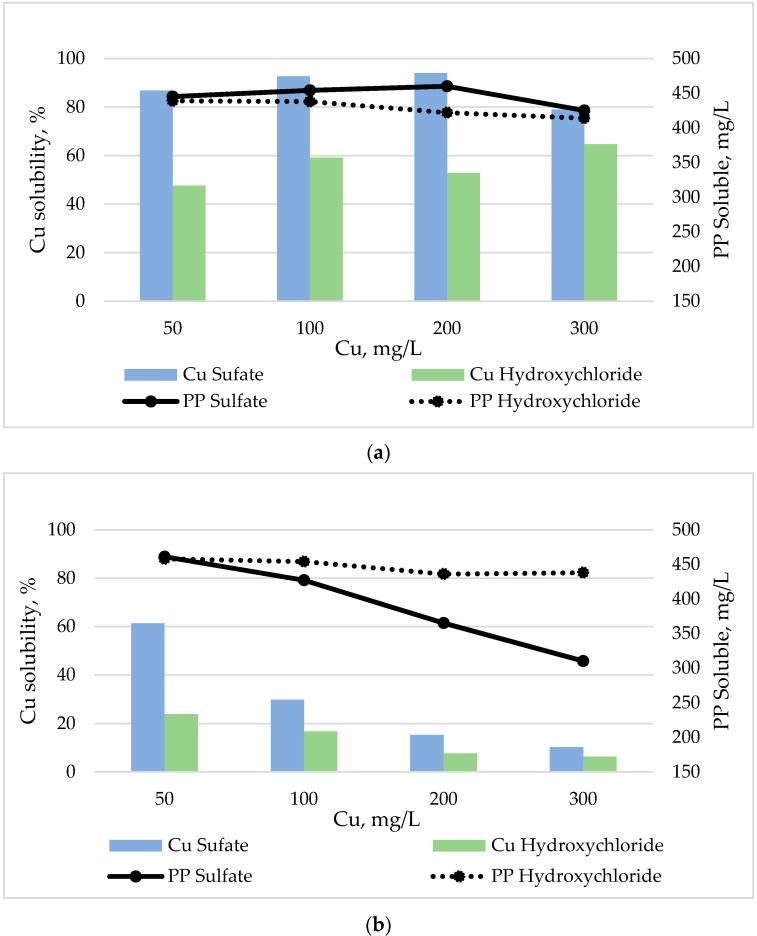
Effects of Cu source and level on Cu solubility and soluble phytic phosphorus (PP) content at pH 4.5 (**a**) and 6.5 (**b**) in the presence of phytic acid.

**Table 1 animals-10-01133-t001:** Composition of the basal diet for Cu preference experiments, as-fed basis.

Ingredients, %	Basal Diet
Wheat	30.00
Maize	23.23
Soybean meal 47% Crude protein	13.85
Sweet milk whey	11.07
Barley	9.00
Soybean meal heat treated	4.00
Fishmeal	2.50
Lard	1.80
Porcine plasma	1.50
Calcium carbonate	0.73
Mono-calcium phosphate	0.25
L-Lysine 50	0.74
DL-Methionine	0.22
L-Threonine	0.21
L-Valine	0.13
L-Tryptophan	0.07
Salt	0.30
Vitamin mineral premix ^1^	0.40
Calculated composition, %	
Dry matter	89.08
Net energy, kcal/kg	2460
Crude protein	19.22
Neutral detergent fiber	8.47
Ether extract	4.72
Ca	0.60
Total P	0.52
Dig P	0.30
Analysed composition, % ^2^	
Dry matter	91.15
Crude protein	19.55
Neutral detergent fiber	9.68
Ether Extract	4.04
Ash	4.62

^1^ Provided per kg of feed: vitamin A (acetate): 12,000 IU; vitamin D3 (cholecalciferol): 2000 IU; vitamin E: 75 IU; vitamin K3: 2 mg; vitamin B1: 3 mg; vitamin B2: 7 mg; vitamin B6: 7.3 mg; vitamin B12: 0.06 mg; D-pantothenic acid: 17 mg; niacin: 45 mg; biotin: 0.2 mg; folacin: 1.5 mg; Fe (chelate of amino acid): 80 mg; Zn (sulphate pentahydrate): 100 mg; Mn (dimanganese chloride trihydroxide): 45 mg; I (calcium anhydrous): 0.7 mg; Se (sodium): 0.3 mg; butylated hydroxytoluene (BHT): 2 mg. Phytase: 500 FTU (Axtra PHY TPT, Danisco, Marlborough, UK). ^2^ Analyzed Cu content in experimental diets: Cu sulfate pentahydrate (29 mg/kg), Cu sulfate pentahydrate (154 mg/kg) and Cu hydroxychloride (152 mg/kg).

**Table 2 animals-10-01133-t002:** Growth performance of pigs in Cu preference experiments ^1^.

Item	Growth Performance ^2^		
	BW (7 Day), kg	BW (14 Day), kg	ADG, g
Cu dose preference ^3^			
Comparison (a)	6.63 (±1.286)	7.76 (±1.255)	160.9 (±34.80)
Comparison (b)	6.66 (±0.931)	7.81 (±1.034)	165.1 (±63.36)
Cu source preference ^4^	7.08 (±1.078)	8.38 (±1.155)	184.4 (±32.26)

^1^ Data are means of 12 pens with 23 pigs per replicate pen. ^2^ Body weight, BW; Average daily gain, ADG. Values in parenthesis indicate the standard deviation of means.^3^ Dose preference test between Cu sulfate at 15 mg/kg and Cu sulfate at 150 mg/kg (a) and between Cu sulfate at 15 mg/kg and Cu hydroxychloride at 150 mg/kg (b). ^4^ Source preference test of Cu at 150 mg/kg of either Cu sulfate or Cu hydroxychloride.

**Table 3 animals-10-01133-t003:** Feed preference percentage at short-term period (day 7–9), long term period (day 9–14) and for the entire experimental period (day 7–14) for Cu preference experiments ^1^.

Item	Cu Level, mg/kg	Preference Percentage		
		Short-Term	Long-Term	Entire Trial
Cu dose preference ^2^				
Comparison (a)				
Sulfate	15	36.04 *	44.16	41.80 *
Sulfate	150	63.96 *	55.84	58.20
SEM ^4^		4.692	7.985	6.030
*p*-value		0.0004	0.312	0.068
Comparison (b)				
Sulfate	15	23.49 *	47.69	41.12 *
Hydroxychloride	150	76.51 *	52.31	58.88
SEM		5.646	8.351	5.589
*p*-value		0.0001	0.698	0.035
Cu source preference ^3^				
Sulfate	150	37.76 *	43.71	42.71
Hydroxychloride	150	62.24 *	56.29	57.29
SEM		5.456	5.838	4.693
*p*-value		0.004	0.141	0.039

^1^ Data are means of 12 pens with 23 pigs per replicate pen. ^2^ Dose preference test between Cu sulfate at 15 mg/kg and Cu sulfate at 150 mg/kg (a) and between Cu sulfate at 15 mg/kg and Cu hydroxychloride at 150 mg/kg (b). ^3^ Source preference test of Cu at 150 mg/kg of either Cu sulfate or Cu hydroxychloride. ^4^ Standard error of the mean, SEM. * Asterisks indicate that experimental diet is significantly different (*p* < 0.05) to the preference neutral value (50%) using T–test.

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
