# Peer review of "Dietary Preference of Newly Weaned Pigs and Nutrient Interactions According to Copper Levels and Sources with Different Solubility Characteristics"

_animals, 2020, doi:10.3390/ani10071133_

Round 1

Reviewer 1 Report

This article assessed Cu preference of pigs with two experiments, results showed that weaning pigs were able to choose diets with higher Cu levels, and the dietary preference relates to solubilities and chemical differences. The idea was interesting. However, I have observed that several aspects need to be clarified and/or discussed before the paper can be considered suitable for publication. The specific comments are as follows:

Abstract

L31: post–weaning be post-weaning

L41: “its” should be “their”

Introduction:

L 59: “properly” should be “proper”

L 68: “Other” should be “Another”

L 80: “digesta” should be “digested”

Materials and Methods:

L 93: what was the feeding level? Why did you adapt pigs for 7 days? You want to research the Strategies to promote an early feed acceptance and to avoid feed neophobia and rejection after weaning, so, why not just order an experimental diet after weaning?

L 114: “according” should be “according to”

L 127-128: you said the d 7–9 is short–term preference, and from d 9–14 is long–term preference, are there any references?

L 133: delete “an”

L 137: make sure hydroxychloride is right. (or hydroxy chloride, or hydrochloride )

Results:

L 157-158: please change the “is” to are, and “are” to was.

L 211: “ussing” misspelled

L 219: “its” should be “their”

Discussion:

L344-345: no distinctions in growth performance in the present study pigs, too short an experiment??

Some sentences may be considered wordy. Consider changing the wording, such as L281.

Grammar and spellings require careful examination of the manuscript.

Author Response

General comments

Authors: The general comments from reviewer 1 have been properly addressed according the summary review table as follows:

-Introduction: The introduction was improved with a broader description of the study, the contextual background, and the research importance (Now Lines 54-55, Now Lines 57-59 and Now Lines 62-65).

-Methodology: the description of the methodology used was enhanced in order to clarify and attend the reviewer comments and suggestions: a) Animals and Housing (Now Line 96, Now Lines 98-100, Now Lines 101, 103); b) Experimental design (Now Lines 110-111, Now Lines 114-117); c) Experimental procedures (Now Lines 139-144).

-Results: the description of results has been improved to clarify and attend the reviewer comments and suggestions (Now Lines 178-180). The growth performance results are shown in Table 2, in order to provide the growth performance information. It has been previously omitted due to its lack of relevance for the final conclusions. The wording of some sentences has been improved as suggested, including figure captions and legends.

-Discussion and Conclusions: A properly discussion concerning feed preference mechanism has included to remark the importance and understanding of the work (Lines 350-356). Conclusions have been rephrased to be clearer (Now Lines 405-407; Now Lines 410-412).

Comments and Suggestions for Authors

This article assessed Cu preference of pigs with two experiments, results showed that weaning pigs were able to choose diets with higher Cu levels, and the dietary preference relates to solubilities and chemical differences. The idea was interesting. However, I have observed that several aspects need to be clarified and/or discussed before the paper can be considered suitable for publication. The specific comments are as follows.

Authors: Thank you for all constructive and valuable comments. All your comments have been responded in a point-by-point manner by using the "Track
Changes" function in Microsoft Word.

Abstract

L31: post–weaning be post-weaning

Authors: The change has been performed here and in the entire manuscript (Now Line 31)

L41: “its” should be “their”

Authors: The personal pronoun has been amended (Now Line 41).

Introduction:

L 59:properly” should be “proper”

Authors: Corrected. The mistake was amended (Now Line 62).

L 68: “Other” should be “Another”

Authors: Agree. The change has been made (Now Line 72).

L 80: “digesta” should be “digested”

Authors: The sentence was improved to help understating (Now Line 84).

Materials and Methods:

L 93: what was the feeding level? Why did you adapt pigs for 7 days? You want to research the Strategies to promote an early feed acceptance and to avoid feed neophobia and rejection after weaning, so, why not just order an experimental diet after weaning?

Authors: Answering to your first question: if you refer to the diet provided immediately after weaning it was a commercial diet commonly used in this farm with a feeding level of Cu at 9 mg/kg (Line 112). During this first week, feed was provided manually to ensure complete availability to the feed. At the end of the week, pan hopper feeders were emptied and replaced by two new pan hopper feeders in each pen. Both, pan hopper feeders were hand filled to completely ensure free access to both diets in order to avoid biases. This consideration has been also included in the methodology section (Now Lines 110-117).

Regarding the adaptation week, it is necessary to mention that usually newly weaned pigs do not eat large amounts of solid food, especially when they are weaned at a very young age. From the literature it is known that during the first week after weaning all the environmental and physiological changes cause that the pigs do not feed voluntarily, even losing weight. Therefore, to avoid biases in the results, we designed the experiment allowing the pigs an adaptation to the new environment, pen mates and feed. Furthermore, based on the previous experience within our research group, it has been observed that the results can be more reliable allowing this adaptation period, even up to 14 days after weaning, as the experimental design of the work by Blavi et al., 2016. This reflection has been properly addressed in methodology section (Now Lines 110-111).

L 114: “according” should be “according to”

Authors: The mistake was amended (Now Line 125).

L 127-128: you said the d 7–9 is short–term preference, and from d 9–14 is long–term preference, are there any references?

Authors: The reference periods were adapted from the works of Forbes (2010; “Palatability: principles, methodology and practice for farm animals”), Torrallardona and Solà-Oriol, (2009; “Evaluation of free-choice feedstuff preference by pigs”) and Roura et al., (2008; “Unfolding the codes of short-term feed appetence in farm and companion animals. A comparative oronasal nutrient sensing biology review”). The methodology section has been improved considering these references (Now line 139-144).

L 133: delete “an”

Authors: The mistake was amended (Now line 149).

L 137: make sure Hydroxychloride is right. (or hydroxy chloride, or hydrochloride )

Authors: Thanks. We have confirmed that the correct spelled is Hydroxychloride.

Results:

L 157-158: please change the “is” to are, and “are” to was.

Authors: Apologies. The change has been performed (Now Line 172 and Now Line 173).

L 211: “ussing” misspelled

Authors: The word has been written correctly now (Now Line 227).

L 219: “its” should be “their”

 Authors: The sentence has been improved to help understanding (Now Line 236).

Discussion:

L344-345: no distinctions in growth performance in the present study pigs, too short an experiment??

Authors: The experimental design was aimed to assess only the feed preference for two levels of Cu (15 and 150 mg/kg) and for two commercial sources (Sulfate and Hydroxychloride). Therefore, both experimental diets (15 and 150 mg/kg or Sulfate and Hydroxychloride) were provided at the same time and in the same pen. Consequently, no reliable conclusions on the effect of Cu levels or Cu sources on growth performance could be drawn. Nevertheless, some growth performance parameters are shown now in Table 2.

Some sentences may be considered wordy. Consider changing the wording, such as L281.

Authors: The manuscript regarding grammar and spelling redaction has been improved. The wording change has been made in all figures (Now line 273-274, Now Line 297-298, Now Line 313-314).  

Grammar and spellings require careful examination of the manuscript.

Authors: Apologies and thank you for your comments and suggestions. The manuscript has been reviewed by a native English speaker (professor) from our Linguistic Service from Universitat Autònoma de Barcelona. Additionally, one word in the tittle of the work has changed as suggested by the native speaker reviewer (Now Line 3)

Reviewer 2 Report

The paper is very interesting, it deals with a very important topic for a very delicate phase of life of the pig and determined for its future growth and health conditions. The work in addition to contributing to the knowledge on the ability to choose the pig at this stage allows you to design diets that are increasingly suitable for nutritional needs

Author Response

Reviewer 2

General comments

Authors: Thank you for your comments on our work. The manuscript has been improved according the suggestions from reviewer 1 and 3. Moreover, to ensure the quality of the manuscript, it has been reviewed by a native English speaker (professor) from our Linguistic Service from Universitat Autònoma de Barcelona.

Comments and Suggestions for Authors

The paper is very interesting, it deals with a very important topic for a very delicate phase of life of the pig and determined for its future growth and health conditions. The work in addition to contributing to the knowledge on the ability to choose the pig at this stage allows you to design diets that are increasingly suitable for nutritional needs

Authors: Thank you very much. We really appreciated your time, comments and guidance.

Reviewer 3 Report

Dear Authors

The article submitted for review, entitled: "Dietary Preference of Newly Weaned Pigs and Nutrient Interactions by Copper Levels and Sources with Different Solubility Characteristics" raises an important issue related to the nutrition of weaned piglets in the context of copper (Cu) content and origin.

The presented article describes the results obtained from a very representative number of animals (n = 828). In addition, the experiment should be considered as properly planned and well conducted. The presented results are clear and correctly illustrated.

However, from the reviewer's point of view I must point out the weaknesses of the article.

The authors indicate that the aim of the work was:

“the aim of the study was to evaluate the preference of weaned pigs for diets that contain added Cu at 15 mg/kg (nutritional) and 150 mg/kg (high) as inclusion levels (dose preference study, Experiment 1) from either Cu sulfate or Cu hydroxychloride source (source preference study, Experiment 2)” (line: 75 – 78)

and

“in vitro assay was performed to determine trace mineral properties in similar conditions to those found in the oral cavity and digesta by testing in vitro Cu solubility of each source  (at 50, 100, 200 and 300 mg Cu/L) and its interaction with phytic acid, in buffer solutions at pH 2.5, 82 4.5 and 6.5” (line 79 – 82)

Turning to the discussion chapter, the authors themselves indicate publications in which these aims were achieved and described, e.g.

  1. Coble, K.F.; Card, K.N.; DeRouchey, J.M.; Tokach, M.D.; Woodworth, J.C.; Goodband, R.D.; Dritz, S.S.; Usry, J. Influence of copper sulfate and tribasic copper chloride on feed intake preference in finishing pigs. Kansas Agric. Exp. Stn. Res. Reports 2013, 181–185.
  2. van Kuijk, S.J.A.; Fleuren, M.A.; Balemans, A.P.J.; Han, Y. Weaned piglets prefer feed with hydroxychloride trace minerals to feed with sulfate minerals. Transl. Anim. Sci. 2019, 3, 709–716.
  3. Pang, Y.; Applegate, T.J. Effects of copper source and concentration on in vitro phytate phosphorus hydrolysis by phytase. J. Agric. Food Chem. 2006, 54, 1792–1796

Presented situation forces me to state that, the article is not an original work but a reproductive one. Furthermore, described conclusions already have been described and are well known.

It is also worth noting that, the authors in the “Discussion” chapter indicate that they did not find differences in productivity. First of all, "Discussion" should not be a place to present the result (no changes is also a result)(line 344-345). Secondly, in the "Results" chapter should be given parameters such as: ADFI, BW, daily body weight gain, etc.

The authors did not perform a test of serum Cu concentration (which they themselves stated in the discussion chapter – line 325-326). This result could indicate a difference in copper retention depending on the dose and source.

Suggestions for the future, that may increase the scientific value of the article:

Determination of copper level in faeces. The analysis would allow for a relative assessment of bio-retention in the body.

Knowing that Cu has antibacterial properties, the authors should check the effects on the gastrointestinal microflora.

In conclusion, the lack of originality and limited methodology are the reason that I suggest to reject the submitted manuscript for publication in "Animals".

Author Response

Reviewer 3

General comments

Authors: The general comments from reviewer 3 has been properly addressed according the summary review table as follows:

-Introduction was improved with all the relevant literature and with a broader description of the research importance (Now Lines 54-55, Now Lines 57-59 and Now Lines 62-65).

-Methodology: the description of the experimental procedures and methodology criteria was improved in provide detailed information: a) Animals and Housing (Now Line 96, Now Lines 98-100, Now Lines 101, 103); b) Experimental design (Now Lines 110-111, Now Lines 114-117); c) Experimental procedures (Now Lines 139-144).

Regarding the research design, we consider that the study has a properly design to test the pig preference. It has been designed according the considerations of early preference works (Forbes, 2010: “Palatability: principles, methodology and practice for farm animals”; Torrallardona and Solà-Oriol, 2009: “Evaluation of free-choice feedstuff preference by pigs” and Roura et al., 2008: “Unfolding the codes of short-term feed appetence in farm and companion animals. A comparative oronasal nutrient sensing biology review”).

-Results: the description of results has been enhanced to help understanding (Lines 178-180). Since the two experimental diets were provided at the same time and in the same pen, the growth performance results were omitted due to its lack of relevance for the final conclusions. Nevertheless, some growth performance parameters are now included in Table 2.

-Discussion and Conclusions: A properly discussion regarding feed preference mechanism has been included to remark the importance and understanding of the work (Now Lines 350-356). Conclusions have been rephrased to be clearer (Now Lines 405-407; Now Lines 410-412).

Additionally, to improve the quality of the manuscript, it has been reviewed by a native English speaker (professor) from our Linguistic Service from Universitat Autònoma de Barcelona.

Comments and Suggestions for Authors

Dear Authors

The article submitted for review, entitled: "Dietary Preference of Newly Weaned Pigs and Nutrient Interactions by Copper Levels and Sources with Different Solubility Characteristics" raises an important issue related to the nutrition of weaned piglets in the context of copper (Cu) content and origin. The presented article describes the results obtained from a very representative number of animals (n = 828). In addition, the experiment should be considered as properly planned and well conducted. The presented results are clear and correctly illustrated. However, from the reviewer's point of view I must point out the weaknesses of the article. The authors indicate that the aim of the work was: “the aim of the study was to evaluate the preference of weaned pigs for diets that contain added Cu at 15 mg/kg (nutritional) and 150 mg/kg (high) as inclusion levels (dose preference study, Experiment 1) from either Cu Sulfate or Cu Hydroxychloride source (source preference study, Experiment 2)” (line: 75 – 78) and “in vitro assay was performed to determine trace mineral properties in similar conditions to those found in the oral cavity and digesta by testing in vitro Cu solubility of each source  (at 50, 100, 200 and 300 mg Cu/L) and its interaction with phytic acid, in buffer solutions at pH 2.5, 82 4.5 and 6.5” (line 79 – 82). Turning to the discussion chapter, the authors themselves indicate publications in which these aims were achieved and described, e.g.

  1. Coble, K.F.; Card, K.N.; DeRouchey, J.M.; Tokach, M.D.; Woodworth, J.C.; Goodband, R.D.; Dritz, S.S.; Usry, J. Influence of copper Sulfate and tribasic copper chloride on feed intake preference in finishing pigs. Kansas Agric. Exp. Stn. Res. Reports 2013, 181–185.
  2. van Kuijk, S.J.A.; Fleuren, M.A.; Balemans, A.P.J.; Han, Y. Weaned piglets prefer feed with Hydroxychloride trace minerals to feed with Sulfate minerals. Transl. Anim. Sci. 2019, 3, 709–716.
  3. Pang, Y.; Applegate, T.J. Effects of copper source and concentration on in vitro phytate phosphorus hydrolysis by phytase. J. Agric. Food Chem. 2006, 54, 1792–1796

Presented situation forces me to state that, the article is not an original work but a reproductive one. Furthermore, described conclusions already have been described and are well known.

Authors: Thank you for your time and valuable recommendations. We must remark that the value of the present work relies on the methodology used to evaluate the feed preference and the assessment of the same sources and levels through an in vitro assay. Together, the two works were aimed to explain the possible reasons behind the preference values reported for different Cu levels (15 vs 150 mg/kg) and sources (Sulfate vs Hydroxychloride).

The experimental design and purpose of our work differs from the  previous mentioned studies (References 23, 24, 25). For example: the design, animals age and the preference results reported by Coble et al. (2013) between Cu levels meaningfully differ from the results obtained in our study and which are properly discussed (Lines 361-362). Likewise, the experimental design of the work performed by van Kuijk et al. (2019) differs from our study, especially since in our work besides to focusing on the comparison between Cu sources, we also evaluated the pig preference between nutritional and high Cu levels within the same pen. In addition, it should be noted that the work of Pang and Applegate (2006) was performed 14 years ago and that certainly the commercial Cu sources have changed at least for the tested source (Hydroxychloride), which technological process has been improved in the last years. Since both sources are commonly used in the commercial practice, it is important to test these feed ingredients.  

The main contribution of our results to the scientific and applied nutrition is to provide one of the physiological reasons behind the greater preference of weaned pigs for diets supplemented with a high level of Cu compared to diets with a nutritional or low level (recommended by Animal Nutrition Institutes). The reason to prefer diets with high levels of Cu is the need of animals to restore the Cu deficit left by weaning and probably boosted by the low consumption of solid feed after weaning. Certainly, this has not be evaluated before and together with the in vitro work, shows some indications as to why there is also a difference between the preference of mineral sources. Furthermore, our results contribute to applied and scientific knowledge considering the positive role of Cu in the post-weaning phase and that in some countries the Cu regulations are getting stricter (due to the negative effects of pharmacological levels of Cu on antimicrobial resistances in livestock and humans health). This mean that feeding with this low or nutritional Cu levels, as pointed out by NRC or other nutritional institutions such as FEDNA or INRA, the early feed acceptance, pig’s preference and consequently performance might be negatively compromised. This new knowledge could allow to nutritionist choose and decide Cu levels and sources properly.

It is also worth noting that, the authors in the “Discussion” chapter indicate that they did not find differences in productivity. First of all, "Discussion" should not be a place to present the result (no changes is also a result)(line 344-345). Secondly, in the "Results" chapter should be given parameters such as: ADFI, BW, daily body weight gain, etc.

Authors: Apologies for the reference of results in the discussion section. This phrase was intended to remark that although previous works with different Cu levels have established a positive effect on growth performance, we were not able to drawn conclusion in this area, due to that both experimental diets (15 and 150 mg/kg or Sulfate and Hydroxychloride) were provided at the same time and in the same pen. Consequently, it is not correct to draw conclusions on growth performance of the pigs. Nevertheless, some growth performance parameters are shown in Table 2.

The authors did not perform a test of serum Cu concentration (which they themselves stated in the discussion chapter – line 325-326). This result could indicate a difference in copper retention depending on the dose and source.

Authors: We agree that the effects and differences of Cu vary depending on the dose and source supplied. Indeed, differences in Cu levels and Cu sources were reported in our previous work (Villagómez et al., 2020), with a factorial design of 2×2 with two Cu levels (15 and 160 mg/kg) and two Cu sources (Sulfate and Hydroxychloride). In the mentioned work, we identified differences between Cu levels and sources due to the factorial arrangement.

Since the present work was aimed to evaluate just the pig preference for Cu, the experiment was designed in a different way and considering the experience and reflections of previous works (Forbes, 2010; Torrallardona and Solà-Oriol, 2009; Roura et al., 2008). Therefore, in the present preference study both different diets (15 and 150 mg/kg or Sulfate and Hydroxychloride) were provided at the same time and within the same pen, which disable the attribution of certain blood Cu levels to a specific dietary level or supplemented source in diet. Even if we had analyzed Cu in blood at the end or during the preference experiment, this would be the result of both different diets.  

Determination of copper level in faeces. The analysis would allow for a relative assessment of bio-retention in the body. Knowing that Cu has antibacterial properties, the authors should check the effects on the gastrointestinal microflora.

Authors: We agree about these effects of Cu. However, like in the blood assessment, the same difficulty occurs in evaluating the level of Cu in faeces. The content of Cu in blood, Cu in faeces and the effect of Cu on microbial populations would not contribute as it cannot directly be related to the preference. Only if the Cu preference could be measured individually or daily, the Cu levels founded in blood or faeces could be related to the individual preference. In our previous work (Villagómez et al., 2020), we reported the antibacterial properties of two Cu levels (15 and 160 mg/kg) and two Cu sources (Sulfate and Hydroxychloride) with differences in diversity, evenness and characteristics of Cu levels and sources on microbial populations. Moreover, we were able to assess the presence of antimicrobial resistance gene, considering the likely negative effects of trace minerals in microbial populations.

In conclusion, the lack of originality and limited methodology are the reason that I suggest to reject the submitted manuscript for publication in "Animals".

We sincerely hope that you can reconsider your decision based on the arguments that we have presented and on the scientific and applied value that our work exposes.

Round 2

Reviewer 3 Report

Dear Authors,

Thank you very much for the extensive explanations. Unfortunately, they did not change my doubts about the originality of the work.

If the research, that You carried out, contained additional/innovative data, the scientific value of the manuscript would be much higher. In response to the review, you indicate that You have performed studies on Cu absorption depending on the source and dose (Villagómez et al., 2020). In my opinion, you could combine data from two experiments and create a very high quality article. In my opinion, you have used the "salami slicing" technique to increase the number of publications.

In addition, You indicate that "the commercial Cu sources have changed at least for the tested source (Hydroxychloride), which technological process has been improved in the last years". If such a change took place, why did you not indicate (briefly, in "Introduction" chapter) what was changed in technology and why it is better?

Because of being a reviewer, I have in mind the quality of the "Animals" jurnal. Lack of additional data and reproduction of a known topic, will cause little citation and a decrease in "Animals" IF.

Author Response

Reviewer 3

Comments and Suggestions for Authors

Dear Authors,

Thank you very much for the extensive explanations. Unfortunately, they did not change my doubts about the originality of the work.

If the research, that You carried out, contained additional/innovative data, the scientific value of the manuscript would be much higher. In response to the review, you indicate that You have performed studies on Cu absorption depending on the source and dose (Villagómez et al., 2020). In my opinion, you could combine data from two experiments and create a very high quality article. In my opinion, you have used the "salami slicing" technique to increase the number of publications.

In addition, You indicate that "the commercial Cu sources have changed at least for the tested source (Hydroxychloride), which technological process has been improved in the last years". If such a change took place, why did you not indicate (briefly, in "Introduction" chapter) what was changed in technology and why it is better?

Because of being a reviewer, I have in mind the quality of the "Animals" jurnal. Lack of additional data and reproduction of a known topic, will cause little citation and a decrease in "Animals" IF.

Authors: Dear Reviewer, after a carefully analysis of your comments we would like to invite you to check our previous work (https://academic.oup.com/jas/article/98/5/skaa117/5818979) and make sure that the experimental design and objectives are complete different with the present study. Therefore, we must reject your unsubstantiated judgment about the “salami slicing technique” to increase the number of publications. The mentioned study together with the present preference study and two other studies are part of the doctoral project of the PhD student Sandra Villagómez (the 1st author) rather than a split of information to increase the number of publications.

We must also remark that the originality and value of the present work rely on the methodology used. In Fact, our methodology markedly differs from previous works mainly due to the experimental design, experimental procedures, measurements as well as the higher number of newly piglets tested. Moreover, the levels tested were selected based on the current permitted Cu level in EU (150 mg/kg) for weaned piglets and the low or nutritional Cu level close to that  recommended by NRC, FEDNA, INRA nutritional institutions. In swine industry and nutrition, it is important to evaluate alternatives that allow stimulating an early consumption of feed by the piglet, helping it to rapidly overcome this stressful and stunted phase after weaning. The results pointed out by the present study showed that newly weaned piglets preferred diets supplemented with 150 mg/Cu kg rather than those with nutritional (15 mg/Cu kg) levels. As discussed in our work, the likely reason for pigs preferred diets with high levels of Cu is the need of animals to restore the Cu deficit left by weaning and which is probably boosted by the low consumption of solid feed after weaning. Indeed, this is the first time that these Cu levels are tested at the same time and in the same pen with weaned piglets and together with the in vitro work, shows some indications as to why there is also a difference in between the mineral sources’ preference. This new knowledge could allow to nutritionist choose and decide Cu levels and sources properly with a scientific base. Regarding, the technological process of the hydroxychloride source,  as you could understand a detailed information is impossible to give due to it is part of the know-how of the company.

Based on our experience (several publications) within this area of pig feed preference and palatability, we consider that our work has the sufficient quality to be published.